# Self-Rated Health and Mortality Among Older Adults in Israel: A Comparison Between Jewish and Arab Populations

**DOI:** 10.3390/jcm13226978

**Published:** 2024-11-20

**Authors:** Itamar Shafran, Yael Benyamini, Lital Keinan-Boker, Yariv Gerber

**Affiliations:** 1Department of Epidemiology and Preventive Medicine, School of Public Health, Faculty of Medical & Health Sciences, Tel Aviv University, Tel Aviv 6997801, Israel; 2Bob Shapell School of Social Work, Tel Aviv University, Tel Aviv 6997801, Israel; benyael@tauex.tau.ac.il; 3Israel Center for Disease Control, Ministry of Health, Chaim Sheba Medical Center, Tel-Hashomer 5262000, Israel; lital.keinan2@moh.gov.il

**Keywords:** self-rated health, geriatric, mortality, ethnicity, Israel

## Abstract

**Background**: Self-rated health (SRH) has been shown to predict mortality across a lifespan. However, its predictive value might differ between populations. We compared the association between SRH and mortality in Israeli Jewish and Arab older adults (65+). **Methods**: A prospective cohort study was conducted among Jewish (n = 1463) and Arab (n = 298) participants in the first National Health and Nutrition Survey of Older Adults (2005–2006). SRH was measured on a four-point scale. Mortality data were available from baseline (2005–2006) through 2019. A survival analysis was performed using Cox models. **Results**: Mean baseline age (SD) was 75 (6) years among Jewish participants (54% women) and 72 (5) years among Arab participants (50% women). Jewish participants were more likely to rate their health as not good (35% vs. 29%) or poor (11% vs. 8%) than Arab participants (*p* = 0.01). During a median follow-up of 13.3 years, 896 deaths occurred; 744 in the Jewish group (mean age [SD] 77.8 [6.6] years) and 152 in the Arab group (mean age [SD] 74.0 [5.2] years). The age- and sex-adjusted hazard ratio (HR) for mortality in the Arab vs. Jewish participants was 1.33 (95% CI: 1.12–1.60). Mortality risk increased with declining SRH, with multivariable-adjusted HRs in the lowest vs. most-favorable SRH categories of 2.46 (95% CI: 1.66–3.63) in the Jewish sample and 2.60 (95% CI: 0.98–6.93) in the Arab sample. **Conclusions**: Although Jewish participants reported poorer SRH, their survival rate was better than Arab participants. Lower SRH was consistently and strongly associated with higher mortality in both groups in a dose–response manner.

## 1. Introduction

Subjective health measurements have aroused increasing interest among researchers and healthcare providers because of their value in assessing treatment outcomes and predicting morbidity and mortality. Self-rated health (SRH) is a measure that grades the subjective assessment of one’s health on a scale of four or five levels, ranging from “excellent” to “poor”, and its use has been documented in studies from the 1950s [1]. The first to describe the predictive value of SRH for mortality in the older adult population (aged 65 years and above) were Mossey and Shapiro in 1982 [2]. Subsequently, many studies have examined its predictive value in various cohorts from countries in North America, Europe, Asia, and Australia [3,4]. The predictive value of SRH has also been documented regarding other health outcomes, including rehabilitation after cardiac events, physician services usage, medication use, and hospitalization [5]. Despite adjusting for various objective determinants, including chronic diseases and socioeconomic status (SES), SRH often remains an independent outcome predictor. The predictive value of SRH was demonstrated in both short- and long-term follow-up periods [6,7]. Different hypotheses proposed that many clinical and sociodemographic factors contribute to one’s self-health assessment, making this simple measure an efficient predictive tool for various health-related domains and across time [3]. For example, in a study of post-myocardial-infarction (MI) patients, SRH before the MI and subsequent change in the rating after the MI were strongly associated with mortality [8]. In line with many other studies, the authors concluded that SRH is a comprehensive indicator of health, integrating biological, psychological, and social factors.

Studies have also investigated the heterogeneity in the relationship between SRH and mortality in various subgroups according to demographic characteristics such as sex and age. Regarding gender, studies have shown that SRH predicted mortality more strongly, or only, in men than in women [9,10]. For example, using the National Health and Nutrition Examination Survey (NHANES-I) data (1971–84), Idler and Angel found that SRH predicted mortality only in men but not women. Stratified by age group (45–64 years, 64–75 years), SRH was predictive only in middle-aged men [11]. In contrast, a study by Benyamini et al. [12] based on the Cross-Sectional and Longitudinal Study (CALAS) data (1989–92) that included Jewish Israelis aged 75 years and above, found that in both men and women, SRH predicted mortality in the elderly (75–84) but not in the very elderly (85–94). A gradual decline in the predictive value of SRH with increasing age was supported in additional studies [13,14]. In contrast, others have found no predictive value variability across age groups [15].

Disparities in health perception have been documented for different ethnicities [16]. Studies that compared the predictive value of SRH for mortality across populations were heterogeneous in their results. For example, Assari et al. [17] found that after adjusting for chronic medical conditions, SRH did not significantly predict mortality among Black American adults in contrast to White adults. Ferraro and Kelley-Moore [18] also reported that SRH was a significantly stronger predictor of mortality among White compared to Black participants in the U.S. Notably, the authors reported that SRH was predictive of mortality in both groups only when SRH was treated as a time-dependent variable. Similarly, researchers reported that although Latinos report lower SRH compared to White Americans, they report higher gradings when questioned on specific health domains, indicating that an ethnic-based reporting bias might exist [19]. Indeed, some studies suggested that SRH represents different aspects of health in diverse populations [17,20]. In Israel, a comparison between Jewish and Arab populations found that their subjective health and its determinants differed. Still, the differences were eliminated in some studies after controlling for socioeconomic factors [21,22]. Several studies have shown that SRH is a valid predictor of morbidity and mortality in older people in Israel [23,24], yet a specific comparison between Jews and Arabs regarding the association between SRH and mortality has not been reported.

In this study, we aimed to compare SRH between Jewish and Arab participants and to assess the factors associated with SRH in each population. In addition, we aimed to assess the mortality rate of both populations and to analyze the association between SRH and mortality. We hypothesized that Arab participants would rate their health more positively than their Jewish counterparts and that different factors would be associated with their SRH, based on studies indicating that Arabs often rate their health more positively than objectively observed measures suggest [21,23]. In addition, based on public reports and previous studies, we speculated that the survival rate would be higher among Jewish participants [24]. Lastly, as the older adult population in Israel grows and various sociodemographic factors differ between Jewish and Arab populations, we hypothesized that the association between SRH and mortality would differ between these groups. Specifically, we postulated a stronger association in the Jewish group consistent with other studies demonstrating stronger associations found in majority groups [17]. This study, therefore, aimed to compare SRH and its associated factors, assess mortality rates, and examine the association between SRH and all-cause mortality among Jewish and Arab participants in a nationwide cohort of community-dwelling adults aged 65 years and older.

## 2. Materials and Methods

### 2.1. Study Design

This prospective cohort study was based on the first National Health and Nutrition Survey of Older Adults in Israel (“Mabat Zahav”), conducted between 2005 and 2006 by the Israel Center for Disease Control and the Nutrition Department of the Israel Ministry of Health. At the beginning of the study, participants self-reported various socioeconomic and clinical factors, and anthropometric measurements such as weight and height were taken. Participants were subsequently followed for all-cause mortality until June 2019. The survey included 1852 randomly sampled community-dwelling participants (1536 Jewish and 316 Arab), aged 65 years and above, who were members of the two major health maintenance organizations in Israel (Clalit Health Services and Maccabi Health Services) that at the time of the sampling represented 86% of the older adult population in Israel. Because the percentage of older Arab adults in society was relatively low (6%), the Arab participants were oversampled to ensure a sufficient sample size for the study. To evaluate the differences between the Jewish and Arab populations more precisely, 37 participants categorized as “Non-Arab Christians” and “Other” minorities were excluded from the present analysis. Data were obtained through a personal interview based on a structured questionnaire at the participant’s home in their native language. SRH results were obtained at the beginning of the study when participants were asked to grade their health on a 4-point scale, ranging from 1, “very good” to 4, “poor” [25]. The analysis included 1761 participants with an available SRH assessment and survival information.

### 2.2. All-Cause Mortality

The entire cohort was matched via their national identification number to the Population Registry and followed for all-cause mortality (last updated June 2019) [26].

### 2.3. Socioeconomic Measurements

Family status, years of schooling, monthly household income (categorized into three levels, up to ILS 3484, ILS 3485–6974, and above ILS 6975 based on the original survey cut-off points), alcohol consumption, and religious level (secular, traditional, or religious) were self-reported by the study participants at baseline. The Israel Central Bureau of Statistics estimated neighborhood SES on a 20-point scale based on the 2008 national census [27]. In this study, we used it as a continuous scale, according to residential locations.

### 2.4. Baseline Clinical Measurements

Cognitive function was assessed with the Mini-Mental State Exam (MMSE), which consists of 30 questions and has a maximum score of 30 points [28]. Participants with a score below 17 were excluded from the original “Mabat Zahav” study due to cognitive impairment. Function was assessed using the Katz scale of activities of daily living (ADL) [29] with a maximum score of 15 points, where a score above 11 indicated severe functional limitations and a score of 5 indicated no limitation. Physical activity levels were assessed based on a questionnaire [30,31] and categorized into sufficiently active (more than 150 min of moderate-intensity physical activity, 75 min of high-intensity physical activity, or any equivalent of combined moderate- and high-intensity physical activity), insufficiently active (less than the previous thresholds), or not active (none or less than one activity per week). Chronic diseases [MI, heart failure, other heart disease, stroke or cerebrovascular accident (CVA), cataract, glaucoma, chronic renal failure (CRF), cancer, Alzheimer’s disease, Parkinson’s disease, asthma, other lung disease, diabetes, osteoporosis, hypercholesterolemia, hypertension] were self-reported and summed to provide the total number of conditions. Smoking status, defined as current, past, or non-smoker, was self-reported at the beginning of the study.

### 2.5. Statistical Analysis

Baseline characteristics between the Jewish and Arab groups were compared using standardized mean differences (SMDs). To assess the factors associated with SRH in each ethnic group, we compared clinical and social characteristics between the two extreme categories of “very good” and “poor” SRH using SMDs. In both comparisons, we considered an SMD greater than 1 as a substantial difference, between 1 and 0.5 as moderate, and under 0.5 as small.

The hazard ratio (HR) and 95% confidence interval (CI) for mortality in the Arab vs. Jewish participants were estimated using a Cox regression model, adjusted for age and sex. Participants who survived until the end of follow-up period were censored. Age- and sex-adjusted mortality rates per 1000 person-years in the Jewish and Arab participants stratified by SRH grading were estimated using Poisson regression models.

We also used Cox regression models to assess the HRs (95% CIs) for mortality associated with SRH grading in the entire cohort and by ethnic group. Three models were examined, with hierarchical adjustment for (1) age and sex; (2) socioeconomic variables, including neighborhood SES, education years, and monthly household income; and (3) clinical variables, including ADL score, MMSE score, and the number of chronic conditions. The proportional hazards assumption was tested using the Schoenfeld residuals, with no violations found in any of the models. We assessed the interaction between SRH and ethnicity by adding an SRH-by-ethnicity term to the regression models. The added value of SRH to the discriminatory ability of the models was evaluated by the Harrell’s c-statistic. Analyses were performed using R software, version 4.0.4 (The R Foundation for Statistical Computing, Kaysville, UT, USA), and IBM SPSS Statistics, version 27 (IBM SPSS Inc., Armonk, NY, USA).

## 3. Results

The analysis included 1761 participants, of whom 1463 (83.1%) were of Jewish ethnicity (53.5% women), and 298 (16.9%) were of Arab ethnicity (50.3% women). Participant characteristics by ethnicity group are presented in Table 1. Substantial differences were noted in education years and neighborhood SES, both higher among the Jewish participants. A moderate difference existed in mean age, 75.2 (SD ± 6.4) years in the Jewish and 72.2 (SD ± 4.8) years in the Arab groups. The former also showed a higher household income, lower religiosity, and higher physical activity level. Other notable differences were that Arab participants had lower MMSE scores, were more likely to smoke, and reported fewer chronic conditions.

### 3.1. SRH Gradings

Regarding SRH, while similar proportions (9–10%) in both ethnic groups graded their health as “very good”, 518 (35.4%) and 166 (11.3%) Jewish participants graded their health as “not so good” and “poor”, compared to 85 (28.5%) and 25 (8.4%) among Arab participants, respectively (*p* = 0.01) (Figure 1).

### 3.2. Clinical and Social Factors Associated with SRH

To better understand the factors associated with SRH in each ethnic group, we compared baseline characteristics between the “very good” and “poor” SRH categories by ethnicity (see Table 2). Substantial differences were noted in several clinical and social variables. Female sex, not being married, a lower income, and a lower MMSE score were more strongly associated with poor SRH in the Arab vs. Jewish groups. In contrast, lower education and neighborhood SES were more substantially associated with poor SRH in the Jewish group. Also, contrary to the Jewish group, where participants with poor SRH were more likely to be current smokers, in the Arab group, participants with very good SRH were more likely to be current smokers. Older age, multimorbidity, lower alcohol consumption, and a sedentary lifestyle were similarly associated with poor SRH in both ethnic groups.

### 3.3. Survival by Ethnicity

During a median follow-up time of 13.3 (IQR: 13.1; 13.6) years (among survivors), a total of 896 deaths occurred, 744 (50.9%) in the Jewish group [mean age 77.8 [6.6] years] and 152 (51.0%) in the Arab group [mean age (SD) 74.0 (5.2) years]. The age- and sex-adjusted HR for mortality in the Arab vs. Jewish participants was 1.33 (95% CI: 1.12–1.60, *p* = 0.002; Figure 2).

### 3.4. Mortality Risk Associated with SRH

Mortality rates (per 1000 person-years), adjusted for age and sex, were calculated across SRH categories by ethnic group (Table 3). In both ethnicities, mortality rates increased with decreasing SRH. In the “very good” category, the mortality rate in the Jewish group (29.55, 95% CI 20.97–38.13) was similar to the Arab group (29.55, 95% CI 9.84–49.25). However, in lower SRH categories, the mortality rates were higher in the Arab group, particularly in the intermediate categories.

HRs for mortality associated with SRH were assessed and adjusted for different covariates (Table 4). In the demographic-only model, HRs in the “poor” vs. “very good” categories were 3.36 and 4.36 in the Jewish and Arab groups, respectively. After adjusting for socioeconomic variables, the respective HRs were 3.32 and 4.44. Finally, in the full model that included sociodemographic and clinical covariates, the HRs were 2.46 and 2.60, respectively. A dose–response relationship was noted in all models, where the HR for mortality increased with decreasing SRH. Although the association between SRH and mortality was numerically stronger in Arab vs. Jewish participants, the interaction between SRH and ethnicity was tested in all three models and was found to be non-significant. In both ethnic groups and regardless of the model, the addition of SRH increased the model’s concordance. Still, the differences were significant only in the Jewish group, probably due to differential statistical power (Table 4).

## 4. Discussion

In this study, we used data from a representative population-based cohort of Jewish and Arab older Israeli adults with long-term follow-up to examine the relationship between SRH and mortality. We compared the SRH–mortality association between the two ethnic groups and assessed the potential modifying effect of ethnicity.

### 4.1. SRH Gradings

The results showed significant differences in SRH between Jewish and Arab participants, where Jewish participants rated their health as poorer than Arab participants. This might be partly attributable to the slightly older age of the Jewish group compared with the Arab group, as was also seen in other studies where older participants tended to rate their health as poorer than younger ones [32]. Additionally, SRH seems to be age-adjusted, even when respondents are not explicitly asked to compare to age peers [33]. As both populations mostly live in separate societies and localities, the ratings may reflect differences in the health of the reference group that the participants used to rate their health. Moreover, education and cultural differences likely also contributed to the different ratings. For example, in a study that compared Jewish and Arab adults (25–64 years old) in Israel, the authors suggested that as the Arab population is more conservative, they might have been less open to discussing their health status with the interviewer, compared to the Jewish participants who are more open to discussing their poor health with others. The authors also suggested that the Arab population might avoid talking about health issues due to a cultural aversion to “bad omens” [17,23]. Cultural differences that might affect SRH were also suggested in a study on Arab immigrants in the U.S., which showed that the participants integrated psychological wellness and non-illness-related aspects into their ratings. A recent study noted a similar pattern where Jewish participants tended to report poorer SRH compared to Arabs [21].

### 4.2. Clinical and Social Factors Associated with SRH

When assessing factors associated with “poor” vs. “very good” SRH in the two ethnic groups, we found that older age, multimorbidity, lower alcohol consumption, and a sedentary lifestyle were associated with poor SRH in both groups. However, female sex, widowhood or living alone, lower income, and a lower cognitive performance were more strongly associated with poor SRH in Arabs. In contrast, lower education and neighborhood SES were more strongly associated with poor SRH in Jews. Similar associations were also reported in a recent study, associating “poor” SRH with older age, Jewish ethnicity, low income and education, and chronic diseases [21]. In a previous study comparing Jewish and Arab middle-aged adults (age 25–64) in Israel, an association between marriage and better SRH was found among Arabs only [23]. Our study supports this finding in older Israeli adults. Interestingly, among Arab participants, and in contrast to Jewish participants, current smokers were more likely to grade their health as “very good”. Racial differences in smoking-related disease risk perception have been previously suggested [34,35] and may result from limited access to healthcare information and lower educational levels among Arab participants.

### 4.3. Survival by Ethnicity

Interestingly, although the Jewish participants graded their health as poorer than the Arab participants, their long-term survival rate was significantly higher. This disparity can be explained by the fact that, despite universal national health insurance providing extended health coverage to all Israeli citizens by law [36], the Jewish population has greater access to high-quality care and utilizes preventive checkups and tests more often than the Arab population [37,38,39]. This was also discussed in a study on midlife women in Israel that suggested that Arab women are in triple jeopardy due to ethnic discrimination, low SES, and gender roles in a patriarchal society that further prevent them from receiving accessible healthcare [22]. In addition, differences in health behaviors exist [21]. Indeed, we observed higher physical activity and lower smoking rates among Jewish participants.

### 4.4. Mortality Risk Associated with SRH

In both groups, the poorer the participants perceived their health to be, the higher the mortality rates, in line with previous studies [3]. The association between SRH and mortality was numerically stronger in Arab vs. Jewish participants in all models tested. However, the interaction was not statistically significant, possibly because of the small sample size in the Arab group. The ability of SRH to predict mortality even after adjustments for social and clinical factors can be explained by SRH being an indicator of illness severity and not the mere existence of a disease. Indeed, many older adults suffer from chronic illnesses. Therefore, the adjustment for these factors does not necessarily reduce the predictive value of SRH, as the latter might provide additional information regarding the severity of diseases and the impact of co-morbidities. This adds to the understanding that SRH also spans aspects that are subjectively assessed by the individual and are challenging to measure objectively by other tools [3].

A 2017 meta-analysis [40], based on the Consortium on Health and Ageing: Network of Cohorts in Europe and the United States (CHANCES), assessed the relationship between SRH and mortality. The meta-analysis included participants aged 60 years and above and evaluated the relationship adjusted for sociodemographic and clinical factors. It was found that “fair” and “poor” SRH were consistently associated with higher mortality in most studies covering most European countries and the U.S., and consistent in different age and sex groups, education levels, and health categories. In addition, the meta-analysis reported a more robust association between SRH and mortality than those reported in studies on Asian populations. Our study extends these findings and supports the view that SRH is a comprehensive indicator of health, integrating biological, cognitive, and social factors. Furthermore, although shared and unique characteristics were both correlated with SRH in Arab and Jewish participants, the predictive ability of SRH for mortality was overall comparable between the groups, suggesting that the different health perspectives may impact the rating but not its association with mortality. This aligns with findings from previous studies that compared the association between SRH and mortality across different sociocultural contexts [41].

### 4.5. Strengths and Limitations

The main strengths of our study are its prospective nature, length of follow-up (>13 years), and representative sample of Israel’s older adults. This is the first study to compare the association of SRH with mortality between Jewish and Arab older adults. We adjusted our models for multiple sociodemographic and clinical factors, given the rich dataset. Nevertheless, our study holds some limitations. Despite oversampling, the small absolute number of Arab participants may have limited the power of the study to detect significant correlations and differences between the two groups. In addition, many factors were self-reported by the participants, which can potentially lead to misclassification. Another limitation is the lack of additional measurements related to mental health or other specific functional measures that are known to be correlated with SRH [42]. Also, although the questionnaire and interview were held in the participant’s language, the translation might be responsible for some discrepancies, similar to what was reported for questionnaires in Spanish [43].

Our findings provide important insights into the potential differences in SRH between Jewish and Arab individuals. These insights may have implications for prevention strategies targeting both ethnic groups. Future studies may explore additional social and clinical factors that affect SRH in both populations and compare how the predictive value of SRH varies among other minority groups in Israel and elsewhere and in relation to different outcomes.

## 5. Conclusions

In this cohort of older adults in Israel, poor SRH was associated with higher mortality in both ethnic groups—Jewish and Arab populations. Jewish participants reported poorer SRH yet had a higher survival rate compared to their Arab counterparts. This discrepancy can result from social, behavioral, and clinical characteristics unique to each ethnic group and differential access to high-quality healthcare. However, SRH improved mortality risk discrimination in both groups above and beyond multiple sociodemographic and clinical factors, providing additional support for its use in research and clinical practice. Further research is warranted to better understand the factors influencing SRH in the Jewish and Arab older adult populations.

## Figures and Tables

**Figure 1 jcm-13-06978-f001:**
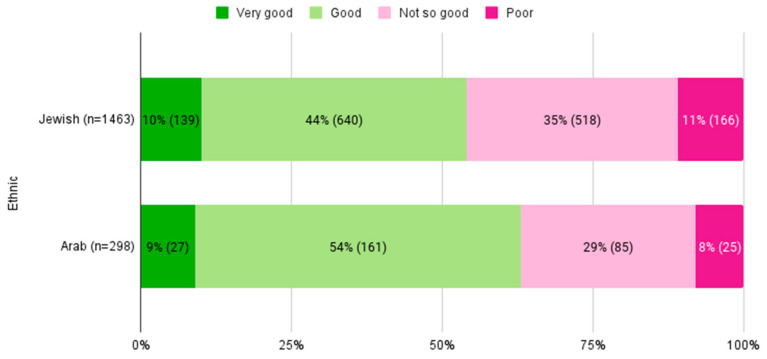
SRH categories by ethnic groups. Percentages (n) of participants in each SRH category by ethnicity. SRH, self-rated health.

**Figure 2 jcm-13-06978-f002:**
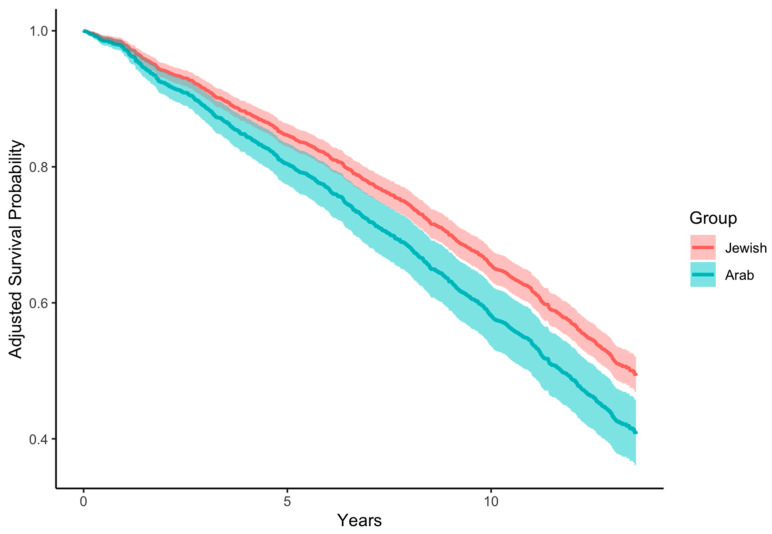
Age- and sex-adjusted survival curve for all-cause mortality in Jewish and Arab participants. Survival curves for all-cause mortality during follow-up (years) for each ethnic group. The area around the curves represents the 95% CI for survival probability.

**Table 1 jcm-13-06978-t001:** Baseline characteristics by ethnic group.

	Jewish	Arab	*p*	SMD
N (%)	1463 (83.1)	298 (16.9)		
Demographic factors				
Age, years (mean (SD))	75.2 (6.4)	72.2 (4.8)	<0.001	0.52
Female sex (%)	782 (53.5)	150 (50.3)	0.36	0.06
Socioeconomic factors				
Married or living with a partner (%)	925 (63.4)	200 (67.1)	0.02	0.08
Education years (mean (SD))	11.5 (4.8)	5.5 (4.6)	<0.001	1.29
Household monthly income (%)			<0.001	0.94
Up to ILS 3484	347 (23.7)	159 (53.4)		
ILS 3485–6974	353 (24.1)	47 (15.8)		
ILS 6975 or more	462 (31.6)	11 (3.7)		
Missing	301 (20.6)	81 (27.2)		
Mean SES by neighborhood in 2008 (SD)	11.5 (3.9)	5.9 (2.3)	<0.001	1.74
Religiosity (%)			<0.001	0.93
Secular	775 (53.0)	52 (17.4)		
Traditional	482 (32.9)	114 (38.3)		
Religious	196 (13.4)	132 (44.3)		
Other/Missing	10 (0.7)	0 (0)		
Clinical factors				
Mean MMSE score (SD)	27.4 (3.0)	25.8 (4.8)	<0.001	0.39
Mean number of chronic conditions (SD)	3.3 (1.9)	2.9 (1.8)	0.002	0.20
Smoking status (%)			<0.001	0.28
Current smoker	136 (9.4)	55 (18.5)		
Former smoker	527 (36.2)	88 (29.5)		
Never smoked	791 (54.4)	155 (52.0)		
Physical activity level (%)			<0.001	0.79
Sufficiently Active	497 (34.0)	50 (16.8)		
Insufficiently Active	455 (31.1)	35 (11.7)		
Not Active	511 (34.9)	213 (71.5)		
ADL (%)			0.67	0.06
No limitations	1135 (78.3)	237 (79.5)		
Mild limitations	260 (17.9)	53 (17.8)		
Severe limitations	54 (3.7)	8 (2.7)		

ADL = activities of daily living, ILS = Israeli Shekel, MMSE = Mini-Mental State Exam, SD = standard deviation, SES = socioeconomic status, SMD = standardized mean differences.

**Table 2 jcm-13-06978-t002:** Comparison of essential characteristics between “very good” and “poor” SRH categories by ethnicity.

	Jewish			Arab		
	Very Good	Poor	SMD	Very Good	Poor	SMD
	139	166		27	25	
Age, years (mean (SD))	73.3 (5.87)	76.5 (6.64)	0.51 −	71.0 (5.06)	73.7 (6.94)	0.44 −
Female sex (%)	62 (44.6)	108 (65.1)	0.42 −	6 (22.2)	16 (64.0)	0.93 −
Single-family status (%)	38 (27.3)	82 (50.0)	0.47 −	3 (11.1)	11 (44.0)	0.79 −
Married or living with a partner (%)	100 (72.5)	82 (50.0)	0.47 +	24 (88.9)	14 (56.0)	0.79 +
Education years (mean (SD))	11.8 (5.5)	9.0 (5.1)	0.54 +	5.9 (5.5)	4.2 (5.0)	0.32 +
Household monthly income, up to ILS 3484 (%)	16 (14.3)	68 (52.7)	0.89 +	9 (42.9)	19 (90.5)	1.17 +
Religious (%)	68 (48.9)	90 (54.2)	0.11 −	23 (85.2)	22 (88.0)	0.08 +
Mean SES by neighborhood in 2008 (SD)	13.1 (3.5)	10.0 (3.7)	0.87 +	6.9 (2.2)	6.5 (3.5)	0.18 +
Mean MMSE score (SD)	28.0 (2.6)	25.1 (4.5)	0.78 +	27.0 (3.7)	21.9 (5.9)	1.04 +
Mean number of chronic diseases (SD)	1.8 (1.1)	5.1 (2.0)	2.03 −	1.6 (1.4)	4.5 (2.1)	1.61 −
Current smoker (%)	11 (7.9)	19 (11.4)	0.12 −	8 (29.6)	4 (16.0)	0.33 +
Alcohol consumption (%)	82 (59.0)	40 (24.1)	0.76 +	9 (33.0)	3 (12.0)	0.53 +
Not physically active (%)	23 (16.5)	114 (68.7)	1.24 −	8 (29.6)	23 (92.0)	1.66 −
Severe or mild ADL limitations (%)	23 (16.8)	39 (23.6)	0.17 −	9 (33.3)	7 (28.0)	0.12 +

+, Higher values in the “very good” SRH category compared to “poor”; −, Higher values in the “poor” SRH category compared to “very good”; ADL, activities of daily living; MMSE, Mini-Mental State Exam; ILS, Israeli Shekel; SES, socioeconomic status; SRH, self-rated health; SD, standard deviation; SMD, standard mean difference.

**Table 3 jcm-13-06978-t003:** Age- and sex-adjusted mortality rates (95% CIs) per 1,000 person-years across SRH categories by ethnicity.

SRH	Jewish	Arab
Very good	29.55 (95% CI: 20.97–38.13)	29.55 (95% CI: 9.84–49.25)
Good	34.70 (95% CI: 30.36–39.05)	49.20 (95% CI: 37.23–61.18)
Not so good	57.23 (95% CI: 50.59–63.88)	84.43 (95% CI: 61.81–107.04)
Poor	88.35 (95% CI: 72.51–104.18)	100.05 (95% CI: 52.23–147.86)

Mortality rates per 1000 person-years were evaluated using the Poisson regression model, with an offset of the log of follow-up time. The age in the model was defined as 75 years old, with 50% females. CI, confidence interval; SRH, self-rated health.

**Table 4 jcm-13-06978-t004:** Hazard ratios (95% confidence intervals) for all-cause mortality associated with self-rated health categories by ethnicity.

Adjusted Model	SRH				*p* for Trend	C-Statistic (SE) Without SRH	C-Statistic (SE) with SRH Added	*p**
	Very Good	Good	Not So Good	Poor				
Demographic								
Jewish (n = 1463)	1 (ref.)	1.20 (0.88–1.64)	2.07 (1.51–2.82)	3.36 (2.39–4.73)	<0.001	0.690 (0.010)	0.726 (0.009)	<0.005
Arab (n = 298)	1 (ref.)	1.82 (0.91–3.67)	3.31 (1.60–6.82)	4.36 (1.89–10.04)	<0.001	0.670 (0.022)	0.692 (0.021)	0.16
+Socioeconomic								
Jewish (n = 1447)	1 (ref.)	1.21 (0.88–1.66)	2.02 (1.47–2.77)	3.32 (2.34–4.73)	<0.001	0.699 (0.010)	0.729 (0.009)	<0.005
Arab (n = 296)	1 (ref.)	1.85 (0.91–3.74)	3.33 (1.60–6.93)	4.44 (1.89–10.43)	<0.001	0.663 (0.022)	0.692 (0.021)	0.05
+Clinical								
Jewish (n = 1383)	1 (ref.)	1.11 (0.80–1.53)	1.71 (1.22–2.38)	2.46 (1.66–3.63)	<0.001	0.718 (0.009)	0.733 (0.009)	<0.005
Arab (n = 286)	1 (ref.)	1.65 (0.79–3.45)	2.42 (1.10–5.30)	2.60 (0.98–6.93)	0.015	0.693 (0.022)	0.703 (0.021)	0.24

*p**: *p*-values for changes in c-statistic. Demographic model: age + sex; socioeconomic model: previous plus education, monthly household income, and neighborhood SES; clinical model: previous plus ADL, MMSE score, and no. of chronic diseases. Interaction SRH*Ethnicity: *p* = 0.73, 0.70, and 0.63 in the demographic, socioeconomic, and clinical models, respectively.

## Data Availability

Anonymized data and materials have been made publicly available at the Israel Ministry of Health website (https://www.health.gov.il/UnitsOffice/ICDC/mabat/Pages/Mabat_Gold.aspx, accessed on 10 November 2024).

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
