# Peer review of "Self-Rated Health and Mortality Among Older Adults in Israel: A Comparison Between Jewish and Arab Populations"

_jcm, 2024, doi:10.3390/jcm13226978_

Round 1
Reviewer 1 Report
Comments and Suggestions for Authors
Comments are in the attached PDF

Reviewer 2 Report
Comments and Suggestions for Authors
Thank you for submitting your manuscript to Clinical Medicine under mdpi. You can see my comments below.
Abstract:
· Well written
Introduction
· Citation error [1-2] instead of [1]-[2]
· Well written
Materials and methods
· Should use past tense
· The study period has been many years ago (between 2005 and 2006), and follow-up until end of June 2019. It is unclear what data were followed and what data were the authors to present. What about the data collected in that period? When were the data of socioeconomic and clinical measurements taken? The authors should clearly explain the procedure.
Results
· Well-reported
Discussion
· Inadequate. The authors should interpret the results step by step and explain the findings, especially between two populations. What are the indications and provide recommendations if any.
Strengths and limitations
· Well-reported
Conclusions
· Can add some more after revising the discussion
References
Need to follow the journal’s requirement.
Reviewer 3 Report
Comments and Suggestions for Authors
First of all, thank you for submitting to JCM.
The author's research is very interesting and has meaningful research value.
Introduction
The introduction is very well written. However, there are 35 references in the introduction alone. There are many duplicate and old studies here. I request many deletions.
Please write the research objectives more specifically.
Adding research hypotheses would make this longitudinal study more valuable.
The methods and results are also well written.
The author wrote the conclusion as follows.
This discrepancy can result from social, behavioral, and clinical characteristics unique to each ethnic group and differential access to high-quality healthcare.
Is this a race issue? Interpretation is more important than analysis of results in this study. This interpretation should be sufficiently referenced based on other studies in the discussion. Therefore, the authors need to discuss not only social, behavioral, and clinical characteristics, but also national health policy, social security system, social stability, national economy, cultural differences, and food culture.
Round 2
Reviewer 2 Report
Comments and Suggestions for Authors
The manuscript has been revised accordingly. I have no more comment. Good work.
Reviewer 3 Report
Comments and Suggestions for Authors
I have no more comments.